# Feed-Forward 3D Gaussian Splatting Compression with Long-Context Modeling

## Abstract

3D Gaussian Splatting (3DGS) has emerged as a revolutionary 3D representation technique, while its substantial data size hinders broader applications. Feed-forward 3DGS compression, which avoids the time-consuming optimization required for per-scene per-train compressors, offers a promising solution for practical deployment. However, existing feed-forward compression methods struggle to model the long-range spatial dependencies, due to the limited receptive field of transform coding network and the inadequate context capacity utilized in entropy models. To address these issues, we propose a novel feed-forward 3DGS compression method that effectively exploits the long contexts. Specifically, we first formulate a large-scale context structure that comprises thousands of Gaussians based on Morton serialization. Then, we design a fine-grained space-channel autoregressive entropy model to fully exploit this expansive context. Furthermore, we develop an attention-based transform coding model to extract informative latent priors by aggregating features from a wide range of neighboring Gaussians. The proposed method yields a $20\times$ compression ratio for 3DGS in a feed-forward inference and achieves state-of-the-art performance among the feed-forward codecs.

## 1 Introduction

With the rapid development of 3D vision and spatial intelligence, the representation of 3D content has become an active research field in both academia and industry. Among various 3D representations, 3D Gaussian Splatting (3DGS) (Kerbl et al., 2023) has become a prominent technique due to its photorealistic illustration quality and real-time rendering capability. It has found a wide range of applications, including simultaneous localization and mapping (SLAM) (Matsuki et al., 2024; Cheng et al., 2025), virtual reality (Zheng et al., 2024; Hu et al., 2024; Qiu et al., 2025), and 3D content digitization (He et al., 2025; Melnik et al., 2025; Bao et al., 2025). However, the superior representation capability of 3DGS relies on a large number of Gaussian primitives, which poses significant challenges for transmission and storage. For example, representing a realistic scene can require storage on the order of several gigabytes (GB). This substantial overhead necessitates effective compression techniques for 3DGS.

Most existing 3DGS compression methods (Fan et al., 2023; Chen et al., 2024; Wang et al., 2024b; Bagdasarian et al., 2025; Ali et al., 2025b) learn a compact representation from the original multi-view images through exhaustive per-scene per-train optimizations. While these approaches achieve impressive compression ratios, the iterative optimization process often takes from minutes to hours, making them impractical for low-latency applications. Moreover, these methods cannot directly encode in-the-wild 3DGS data without referring to the original multi-view images, as the optimization depends heavily on these ground-truth inputs. In contrast, generalizable compressors employ feed-forward neural networks to encode data. This type of codec has been extensively studied in conventional modalities, such as image (Ballé et al., 2018), video (Lu et al., 2019; Li et al., 2021), and point clouds (Gao et al., 2025), but remains largely underexplored in 3DGS compression. To our knowledge, FCGS (Chen et al., 2025a) is the only existing attempt at feed-forward 3DGS compression. This method employs a multi-path entropy model and voxel grid-based context to build the codec, compressing 3DGS attributes through the inference of a trained neural network. It has demonstrated the feasibility of feed-forward 3DGS compression.

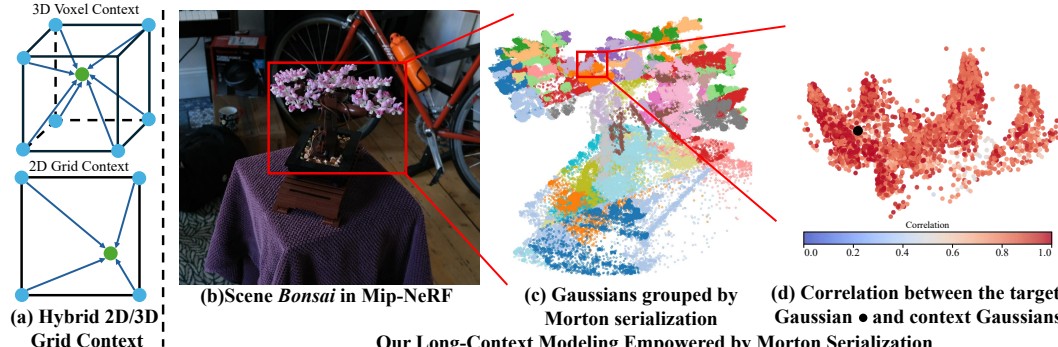

(a) Hybrid 2D/3D Grid Context

(b)Scene *Bonsai* in Mip-NeRF

(c) Gaussians grouped by Morton serialization

(d) Correlation between the target Gaussian ● and context Gaussians

Our Long-Context Modeling Empowered by Morton Serialization

Figure 1. (a) Context illustration of the existing method (Chen et al., 2025a), which utilizes a limited receptive field that includes only several Gaussians. (b) Visualization of the *Bonsai* scene. (c) Visualization of the corresponding 3DGS point cloud, where points are partitioned into Morton-serialized context windows. Points from the same window are painted with the same color. (d) Correlations of Gaussian attributes between a target point and other points within the same context window. Strong correlations persist between distant Gaussians.

However, the design choices of FCGS remain suboptimal regarding the transform coding and context modeling. Specifically, it employs a multi-layer perceptron (MLP) to extract latent features from the original 3DGS, with its entropy model conditioned on a context that combines several neighbors in the voxelized 3D space and 2D projections. The MLP-based transform coding does not effectively eliminate the spatial redundancy since it processes each Gaussian independently and overlooks correlations among different Gaussians. Meanwhile, the expressiveness of this latent representation is limited due to the lack of spatial connectivity. In addition, the voxelized context only incorporates references within a narrow local vicinity, thus failing to capture the long-range dependencies in large-scale scenes. As shown in Fig. 1(d), strong correlations persist across a substantial number of Gaussians. Thus, enhancing the long-range context modeling capabilities of transforms and entropy models is beneficial to boost the compression performance.

To address these challenges, we propose **LocoMoco**, a novel method that achieves **lo**ng-**co**ntext **mo**deling for efficient 3DGS **co**mpression. Our key idea is to scale the receptive field and context capacity to accommodate thousands of Gaussians and use attention mechanisms to effectively capture their long-range dependencies. The pipeline begins by traversing the 3DGS primitives into a 1D sequence via Morton serialization. This sequence is then partitioned into large-scale context windows, where Gaussians within the same window correspond to spatial neighbors, as depicted in Fig. 1(c). Subsequently, self-attention modules correlate Gaussians within each window to learn a compact and informative latent embedding. Then, the large-scale context model estimates the distributions of Gaussian attributes based on a fine-grained space-channel auto-regressive approach. Each window is first evenly split into two groups to establish spatial context, after which channel-wise context is incorporated to exploit correlations across different Gaussian attributes. Finally, space, channel, and latent priors are integrated for precise probability estimation.

To summarize, our contributions are :

- We propose a novel feed-forward 3DGS compression framework that effectively captures long-range dependencies in 3D scenes. To this end, we develop a large-scale context structure tailored for long-context modeling based on Morton serialization.

- We design a serialized attention architecture as the backbone of transform coding network and context model. This design significantly expands the receptive field of the network, which effectively models the long-range dependencies in the extensive context.

- We present a fine-grained space-channel context model for accurate distribution estimation. It partitions the symbol sequence into closely ccorrelated subgroups to fully exploit the dependencies among different Gaussians and different attributes.

- Our method achieves state-of-the-art rate-distortion performance among feed-forward 3DGS compressors, surpassing existing methods with a BD-Rate gain of around 10%.

## 2 RELATED WORK

### 2.1 OPTIMIZATION-BASED 3DGS COMPRESSION

The substantial data volume of 3DGS hinders its wider-spread applications and motivates research interests in 3DGS compression (Bagdasarian et al., 2025; Ali et al., 2025b). Existing 3DGS compression methods primarily adhere to the per-scene per-train paradigm, where a exhaustive optimization is required to yield a compact 3DGS representation based on multi-view image inputs. These methods can be broadly categorized into value-based and structure-based approaches. Value-based methods prune the less contributing elements from the representation (Fan et al., 2023; Lee et al., 2024; Ali et al., 2024; Zhang et al., 2024) and reduce the numerical redundancies using quantization techniques (Girish et al., 2024; Niedermayr et al., 2024; Navaneet et al., 2024b; Ali et al., 2025a). Structure-based methods simplify the 3DGS into self-organized 2D planes (Morgenstern et al., 2023), scaffold grids (Lu et al., 2024), or other more compact data structures (Chen et al., 2024; Wang et al., 2024b; Yang et al., 2024; Liu et al., 2024b;a). Despite notable improvements in compression ratio, the practicality of these methods is limited by the time-consuming per-scene optimization and the reliance on ground-truth multi-view images. In contrast, our work focuses on designing a feed-forward 3DGS compressor suitable for fast and in-the-wild 3DGS compression.

### 2.2 FEED-FORWARD DATA COMPRESSION

Different from the aforementioned methods that require per-scene optimization, most neural image, video, and point cloud compression models work in a feed-forward way. After the training on large-scale datasets, these methods directly compress the data without further optimization. The image compression framework typically comprises of a transform coding model and an entropy model (Ballé et al., 2018). The analysis transform projects an image to a compact latent embedding, while the synthesis transform reconstructs the image from the latent conversely. The entropy model predicts the distributions of the latent elements to reduce the bitrate consumption, based on a context of previously decoded symbols (Minnen et al., 2018; Minnen & Singh, 2020; He et al., 2022; Jiang et al., 2023; Liu et al., 2024c). Video compression models follow a similar pipeline, while features from previous frames are additionally introduced to further enhance the rate-distortion performance (Li et al., 2021; Mentzer et al., 2022; Jiang et al., 2025).

Learning-based point cloud geometry compression methods typically use context models to reduce the bitrate consumption for lossless compression. The orderless point cloud is first transformed into more regular data structures, such as octree and voxel. Then, the octree-based methods predict the occupancy symbols of the octree nodes based on a context of previously decoded ancestor and sibling nodes (Huang et al., 2020; Fu et al., 2022; Song et al., 2023a;b). Similarly, the voxel-based methods estimate the voxel occupancy status based on decoded neighboring and low-resolution voxel grids (Wang et al., 2025a). For attribute compression, hand-crafted or learned transforms are first employed to remove the redundancy within the data (De Queiroz & Chou, 2016; Zhang et al., 2023). Then, a learning-based context model is adopted to further reduce the bitstream length for coding the transform coefficients or latent embeddings (Fang et al., 2022; Wang et al., 2025b).

## 3 PRELIMINARIES

### 3.1 3D GAUSSIAN SPLATTING (3DGS)

3DGS (Kerbl et al., 2023) represents a 3D scene with a set of point-based primitives. Each Gaussian primitive is defined as:

$$g(\boldsymbol{x}) = e^{-\frac{1}{2}(\boldsymbol{x}-\boldsymbol{\mu})^T \boldsymbol{\Sigma}^{-1}(\boldsymbol{x}-\boldsymbol{\mu})}, \tag{1}$$

where $\boldsymbol{\mu} \in \mathbb{R}^3$ and $\boldsymbol{\Sigma} \in \mathbb{R}^{3\times3}$ denote the position vector and the covariance matrix, respectively. The covariance matrix $\boldsymbol{\Sigma}$ can be decomposed into a rotation matrix $\mathbf{R} \in \mathbb{R}^{3\times3}$ and a scaling matrix $\mathbf{S} \in \mathbb{R}^{3\times3}$ as:

$$\boldsymbol{\Sigma} = \mathbf{R}\mathbf{S}\mathbf{S}^T\mathbf{R}^T, \tag{2}$$

where $\mathbf{R}$ and $\mathbf{S}$ are further represented by a quaternion $\boldsymbol{q} \in \mathbb{R}^4$ and a vector $\boldsymbol{s} \in \mathbb{R}^3$, respectively. Besides, each Gaussian contains an opacity property $o \in \mathbb{R}^1$ and a view-dependent color property

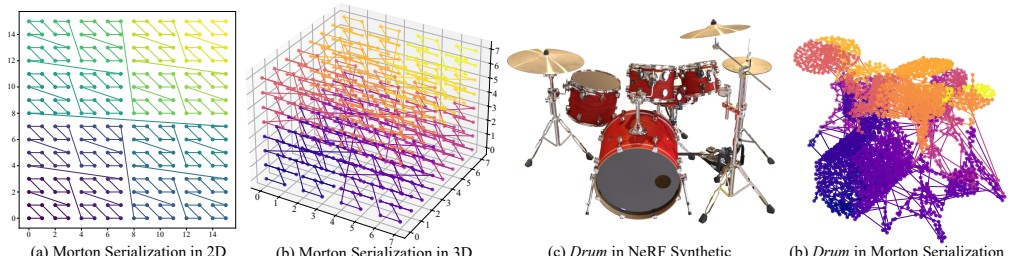

(a) Morton Serialization in 2D   (b) Morton Serialization in 3D   (c) *Drum* in NeRF Synthetic   (b) *Drum* in Morton Serialization

Figure 2. Visualizations of the Morton order in (a) 2D plane and (b) 3D space. We further exhibit the (c) visualization of the *Drum* scene in the NeRF-Synthetic dataset (Mildenhall et al., 2021) and (d) the corresponding 3DGS point cloud, where points are colored according to their indices in the Morton-serialized sequence. It is shown that Morton serialization maintains the spatial proximity.

(i.e., spherical harmonics) $c \in \mathbb{R}^{48}$, which are incorporated into the rendering process as:

$$C = \sum_{i=1}^{N} c_i \alpha_i \prod_{j=1}^{i-1} (1 - \alpha_j). \tag{3}$$

Here, $\alpha_i$ denotes the transmittance of a Gaussian, which is calculated based on its opacity and covariance matrix. As listed above, each 3DGS primitive $g$ requires the position $\mu \in \mathbb{R}^3$, scale $s \in \mathbb{R}^3$, rotation $q \in \mathbb{R}^4$, opacity $o \in \mathbb{R}^1$, and color properties $c \in \mathbb{R}^{48}$ to represent one scene, counting up to 59 parameters in total. Furthermore, faithfully representing a 3D scene requires millions of 3DGS primitives, which leads to heavy storage burdens. As a result, 3DGS compression becomes critical in practical applications.

## 3.2 MORTON CODE

Morton code (Morton, 1966), also known as Morton order or Z-order, is a commonly used technique in data serialization, which functions to map high-dimensional data to one dimension while preserving the neighboring relationship in the high-dimensional space. Given a positive 3DGS coordinate $(x, y, z)$, which is quantized into $d$ bits, the Morton code $\pi$ is generated by interleaving the bits in the binary representation of $x$, $y$, and $z$ as follows:

$$\pi = \sum_{i=0}^{d-1} [\text{Binary}(x, i) << (3i) + \text{Binary}(y, i) << (3i + 1) + \text{Binary}(z, i) << (3i + 2)], \tag{4}$$

where $\text{Binary}(u, i)$ extracts the $i$-th bit of $u$, and "$<<$" stands for the bit shifting operator. The Morton serialization rearranges the Gaussian primitives in the ascending order of Morton codes. Due to their similar Morton codes, neighboring Gaussians in the 3D space are also placed close to each other in the serialized sequence, as visualized in Fig. 2.

## 4 LOCOMOCO: LONG-CONTEXT MODELING FOR 3DGS COMPRESSION

### 4.1 OVERVIEW

3DGS serves as an intermediate for novel view synthesis, and the rendering quality is sensitive to the perturbations on attribute values: minor deviations in Gaussian attributes can lead to substantial distortions in the rendered images. Motivated by previous methods (Chen et al., 2025a), we adopt a hybrid lossy-lossless compression backbone to yield high-fidelity rendering while maintaining a compact data size. We encode the geometric features, including position $\mu$, scale $s$, rotation $r$ and opacity $o$, using lossless compression since the rendering quality is extremely sensitive to these components. Specifically, $\mu$ is firstly quantized into $\hat{\mu}$, which is then compressed using an off-the-shelf point cloud codec G-PCC (MPEG, 2025). The other geometric features $h = \{s, r, o\}$ are discretized using learned quantization steps, and the quantized features $\hat{h} = \{\hat{s}, \hat{r}, \hat{o}\}$ are subsequently encoded via a neural network. As for the color attributes $c$, we first employ a lightweight division network to identify the perturbation-sensitive Gaussians, denoted by $c_{\text{lossless}}$. These Gaussians are then quantized into $\hat{c}_{\text{lossless}}$ and compressed losslessly. Meanwhile, the remaining Gaussians, denoted by $c_{\text{lossy}}$, are encoded via lossy compression for a higher compression ratio.

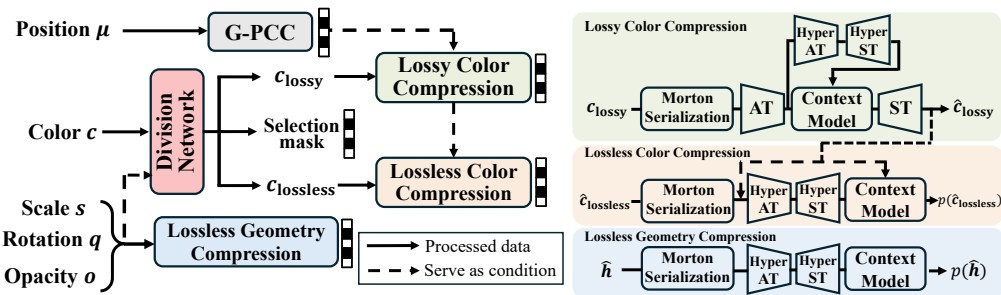

Figure 3. Overview of the proposed LocoMoco. The architectures of each module are illustrated on the right. Here, AT and ST denote the analysis and synthesis transforms, respectively.

The architecture of LocoMoco is illustrated in Fig. 3. It sequentially performs lossless geometry compression, lossless-lossy component division, lossy color compression, and lossless color compression. Each compression module first rearranges the input 3DGS into a 1D Gaussian primitive sequence $G = \{g_1, \cdots, g_N\}$ via Morton serialization. Importantly, the derived sequence $G$ maintains the neighboring relationships in the 3D space, which provides a convenient data structure for retrieving features of spatial neighbors and facilitates our long-context modeling. Then, the derived sequence is partitioned into numerous non-overlapped context windows $x$ of length $L$:

$$G = \{x_1, x_2, \cdots, x_k\}, \quad x_i = \{g_{(i-1)L+1}, \cdots, g_{iL}\}. \tag{5}$$

Here, $k = \lceil N/L \rceil$ is the number of context windows. In particular, the context length $L$ compromises 1024 Gaussians in our implementation. Afterwards, the lossless compression framework extracts the hyper latent embedding $u_i$ from $x_i$, and predicts the probability distribution of $x_i$ based on the quantized latent $\hat{u}_i$ along with the space-channel context. The lossy compression model projects $x_i$ to the latent feature $y_i$, which is compressed by another quantized hyper-prior latent $\hat{z}_i$ and the space-channel context. Then, the model reconstructs Gaussian attributes using the quantized latent $\hat{y}_i$. The transform coding network and entropy model of lossless and lossy share the architecture built on attention blocks.

## 4.2 TRANSFORM CODING

Transform coding projects the Gaussian attributes to a compact latent, where the duplicated patterns are merged. Inspired by 3D representation learning methods (Wu et al., 2024), our LocoMoco generates the latent by applying self-attention on the serialized context window. The structure of the attention block is illustrated in Fig. 4 (Left). It first generates positional encoding for each Gaussian using a dynamic-graph convolutional neural network (DGCNN) (Wang et al., 2019), which extracts the geometric features of the local patch based on the Gaussian positions $\hat{\mu}$. Then, the self-attention layer computes dependencies among Gaussians within the same context window $x_i$, and aggregates features of neighboring Gaussians based on the attention score.

Existing methods use an MLP to learn the latent space, where features of each Gaussian are processed independently (Chen et al., 2025a). This method just combines different features of the same Gaussian primitive and fails to reduce the spatial redundancy by connecting with other Gaussians. The derived features also tend to be less representative, since the independent feature processing cannot analyze the spatial patterns. In contrast, the proposed model learns a spatial-aware latent representation. Note that this self-attention layer facilitates information exchanges between the closely correlated Gaussians. The large context window $x_i$ significantly expands the receptive field, resulting in an informative latent embedding. This enhancement is crucial for lossy compression, as the reconstruction fidelity heavily depends on the quality of the latent representation.

## 4.3 CONTEXT MODEL

LocoMoco adopts a space-channel context model for precise probability density estimation on the serialized sequence. Since the context model predicts different windows independently, without loss of generality, we define the entry of the context model as $m = \{n_1, \cdots, n_L\}$. Here, $n$ corresponds to Gaussian attributes or latent embeddings for lossless and lossy compression, respectively. The sequence $m$ is partitioned into several slices along both spatial and channel dimensions, and different

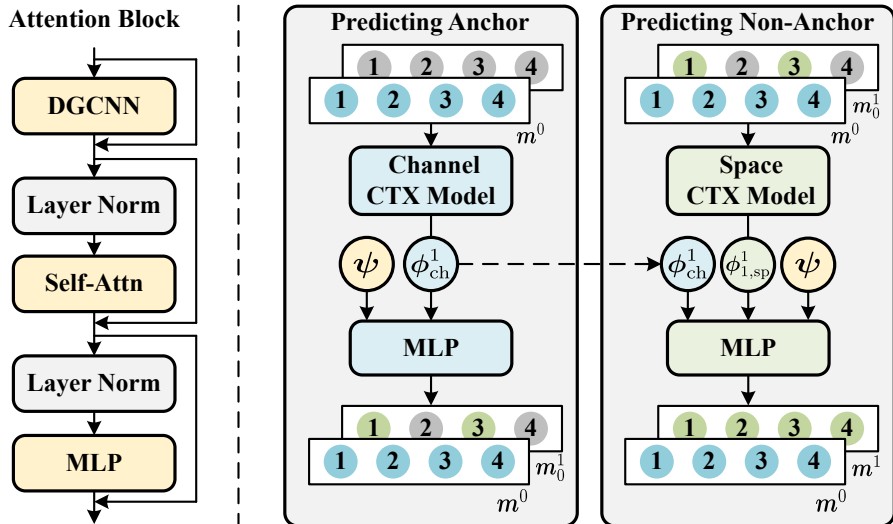

Figure 4. (Left) Architecture of our Morton serialized attention block. (Right) Workflow of the space-channel context model. We illustrate the coding of the subgroup $m_0^1$ and $m_1^1$ based on $m^0$. Here, yellow nodes indicate the latent prior $\psi$, blue nodes denote the channel context $m^0$, and green nodes represent the decoded subgroups in $m^1$.

slices are predicted in serial to conduct fine-grained space-channel context modeling. Specifically, $m$ is first decomposed into two groups spatially:

$$m_0 = \{n_1, n_3, \cdots, n_{L-1}\}, \quad m_1 = \{n_2, n_4, \cdots, n_L\}. \quad (6)$$

This partition extracts elements with even indices and odd indices separately. As the Morton serialization preserves the proximity in 3D space, the anchor $m_0$ can be regarded as the spatial neighbors of the non-anchor $m_1$, which constitutes a strong spatial context. Then, each group is further partitioned into several subgroups along the channel dimension as:

$$m_0^j = \left\{n_1^j, n_3^j, \cdots, n_{L-1}^j\right\}, \quad m_1^j = \left\{n_2^j, n_4^j, \cdots, n_L^j\right\}, \quad (7)$$

where $n_i^j$ indicates $j$-th channel subgroup of $n_i$. The space-channel context model predicts symbol distributions by aggregating the latent features and space-channel context. The corresponding coding pipeline is illustrated in Fig. 4. It first extracts a latent prior $\psi$ from the latent $\hat{u}$ in lossless compression or $\hat{z}$ in lossy compression. Then, it aggregates contextual features from the previously coded subgroups as:

$$\phi_i^j = \sigma_{ep}(\phi_{ch}^j, \phi_{i,sp}^j) \quad \phi_{ch}^j = \sigma_{ch}(m^{<j}), \quad \phi_{i,sp}^j = \sigma_{sp}(m_{<i}^j). \quad (8)$$

Here, $\phi_i^j$ is the space-channel context for $m_i^j$. The channel context model $\sigma_{ch}$ adopts self-attention blocks to extract features from previously coded channel subgroups $m^{<j}$. The space context model $\sigma_{sp}$ is only available for predicting the non-anchor $m_1^j$. It uses self-attention modules to introduce prior features from the decoded $m_0^j$. Specifically, features for anchor positions are derived by concatenating the latent features, channel context, and the embedding of $m_0$. While the features for non-anchor positions are obtained by fusing the latent embeddings and channel context of $m_1$. Finally, the latent prior $\psi$ and space-channel context $\phi_i^j$ are combined to predict the symbol distributions. In addition, we introduce features from the decoded lossy Gaussians to predict the latent of the lossless ones (i.e., $\hat{u}$) via a space context model. Since the attention mechanism can effectively model the long-range dependencies within the large-scale context $m$, the available context capacity of the proposed model has been significantly expanded compared to the voxel-based context (Chen et al., 2025a).

The entropy model uses a discretized logistic mixture (Salimans et al., 2017) to encode $\hat{h}$ and $c_{lossless}$ in lossless compression, while the coding of latent $\hat{y}$ is based on the Gaussian distribution. Besides, the non-parametric density model (Ballé et al., 2018) is adopted to encode $\hat{u}$ and $\hat{z}$.

## 4.4 TRAINING

Our pipeline is optimized using the rate-distortion (RD) objective. The overall training objective is formulated as:

$$\mathcal{L}_{RD} = \mathbb{E}_{\boldsymbol{G}}[D(\delta(\boldsymbol{G}), \delta(\hat{\boldsymbol{G}})) + \lambda(R_{\text{lossless\_geo}} + R_{\text{lossy\_color}} + R_{\text{lossless\_color}})], \tag{9}$$

$$R_{\text{lossless\_geo}} = \mathbb{E}_{\boldsymbol{G}}[\sum_{i,j} - \log p(\hat{\boldsymbol{h}}_i^j | \boldsymbol{\phi}_i^j, \hat{\boldsymbol{u}}_{\text{geo}}) - \log p(\hat{\boldsymbol{u}}_{\text{geo}})], \tag{10}$$

$$R_{\text{lossy\_color}} = \mathbb{E}_{\boldsymbol{G}}[\sum_{i,j} - \log p(\hat{\boldsymbol{y}}_i^j | \boldsymbol{\phi}_i^j, \hat{\boldsymbol{z}}) - \log p(\hat{\boldsymbol{z}})], \tag{11}$$

$$R_{\text{lossless\_color}} = \mathbb{E}_{\boldsymbol{G}}[\sum_{i,j} - \log p(\hat{\boldsymbol{c}}_{i,\text{lossless}}^j | \boldsymbol{\phi}_i^j, \hat{\boldsymbol{u}}_{\text{color}}, \hat{\boldsymbol{c}}_{\text{lossy}}) - \log p(\hat{\boldsymbol{u}}_{\text{color}})]. \tag{12}$$

Here, $\boldsymbol{G}$ is a 3DGS sampled from the dataset, $\hat{\boldsymbol{G}}$ is the recovered 3DGS attributes by merging the results of lossy and lossless compression, $\delta$ indicates the rendering pipeline. The rate term $R$ comprises the bitrate for lossless geometry, lossy color, and lossless color compression. The distortion $D$ is defined as the combined MSE and SSIM loss between the rendered images $\delta(\boldsymbol{G})$ and $\delta(\hat{\boldsymbol{G}})$, which is averaged on six randomly chose views. $\lambda$ is a hyper-parameter that controls the rate-distortion trade-off.

## 5 EXPERIMENTS

### 5.1 EXPERIMENTAL SETUP

**Datasets.** We train the LocoMoco based on the DL3DV-GS dataset (Ling et al., 2024; Chen et al., 2025a), which contains 6770 samples. Each sample comprises a set of multi-view images, corresponding camera poses, and 3DGS attributes. We follow the dataset division of FCGS (Chen et al., 2025a), where 100 scenes are preserved for evaluation while the others are used for training. Aside from these 100 samples, we conduct a comprehensive evaluation on the Mip-NeRF 360 (Barron et al., 2022) and Tanks & Temples (Knapitsch et al., 2017) datasets. We follow the vanilla 3DGS training pipeline (Kerbl et al., 2023) to obtain the 3DGS samples from these datasets.

**Implementation Details.** Our method is implemented based on the Pytorch framework (Paszke et al., 2019), aided with CompressAI (Bégaint et al., 2020) and FlashAttention (Dao et al., 2022) repositories. The latent feature $\hat{\boldsymbol{u}}$ is composed of 8 and 48 channels for lossless geometry and color compression, respectively. For lossy compression, the latent embedding $\hat{\boldsymbol{y}}$ includes 64 channels. Consistent with FCGS, the batch size is set to 1 for training. The parameter $\lambda$ is selected from $[1e-4, 2e-4, 4e-4, 8e-4, 16e-4]$ to switch among different rate-distortion trade-off levels. More details about the model architecture and implementation are presented in Appendix A.

**Baselines.** To the best of our knowledge, FCGS (Chen et al., 2025a) is the only existing exploration of feed-forward 3DGS compression. Therefore, we mainly compare LocoMoco against FCGS. We evaluate the performance of FCGS based on its official codebase and model weights. We also conduct comparisons with various per-scene per-train 3DGS codecs in Appendix B.4.

**Metrics.** The rate-distortion performance is measured using the storage size of the compressed 3DGS (in MB) and peak signal-to-noise (PSNR, in dB). Besides, we use the widely employed Bjøntegaard Delta Rate (BD-Rate) (Bjontegaard, 2001) to compare the compression efficiency of different methods. BD-Rate represents the relative data size savings between different methods under the same reconstruction quality, and thus the lower BD-Rate reflects better compression performance. Further comparisons of the structural similarity index measure (SSIM) and learned perceptual image patch similarity (LPIPS) are provided in Appendix B.1.

### 5.2 RESULTS

**Quantitative Results.** Fig. 5 illustrates the rate-distortion performance of FCGS and LocoMoco on the DL3DV-GS, Mip-NeRF, and Tanks & Temples datasets. Compared to the vanilla 3DGS that

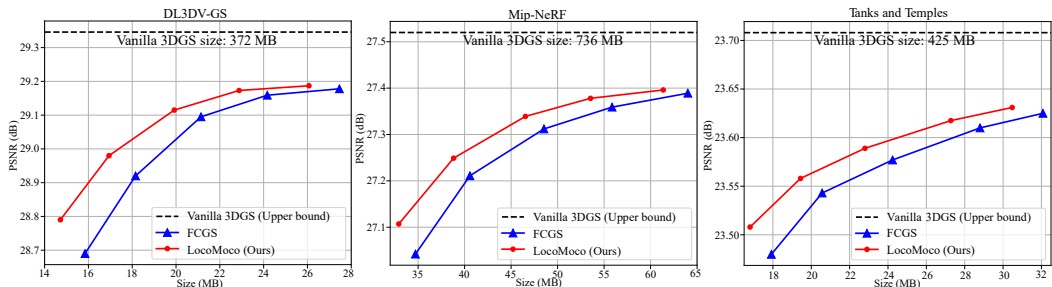

Figure 5. Rate-distortion curves in terms of rendering quality and storage size.

Table 1. (Left) BD-Rate gains relative to FCGS. (Right) Encoding and decoding time efficiency evaluated on DL3DV-GS.

| Method | BD-Rate ($\downarrow$) | | | Time Consumption (s) | |
|---|---|---|---|---|---|
| | DL3DV-GS | Mip-NeRF | Tanks & Temples | Encoding | Decoding |
| FCGS | 0 | 0 | 0 | 15.65 | 11.33 |
| LocoMoco (Ours) | -10.11% | -9.39% | -10.35% | 11.33 | 12.89 |

costs hundreds of MB, our method greatly reduces the storage size and bears minor visual distortion (less than 0.5 dB drop in PSNR). Compared with FCGS, LocoMoco achieves consistently better rate-distortion performance across a wide range of bitrates among various datasets. Furthermore, Table 1 suggests that the proposed method obtains around 10% BD-Rate savings upon FCGS, which is a significant improvement.

The improvements stem from several advantageous designs. The serialized-attention-based transform coding produces a compact and informative latent representation, which reduces storage and improves reconstruction fidelity simultaneously. The fine-grained space-channel context model effectively captures the long-range dependencies within the large-scale scenes, leading to precise probability estimation and reduced bitstream length.

Furthermore, we compare the coding times of FCGS and LocoMoco in Table 1. It is shown that LocoMoco yields notable compression efficiency improvements upon FCGS while maintaining a comparable coding latency. Besides, as a feed-forward compression method, LocoMoco delivers a coding latency at the second level, which appears significantly faster than the time consumption of per-scene per-train compression methods that ranges from minutes to hours.

**Qualitative Results.** In Fig. 6, we visualize the ground truth image, the rendering result of vanilla 3DGS, and the rendered image from the compressed 3DGS produced by FCGS and LocoMoco. The proposed method yields a 26 $\times$ data size reduction, while preserving a high-fidelity reconstruction. In contrast, the rendering result of FCGS exhibits blurred reconstructions and color variations. We present more visualizations in Appendix B.5.

### 5.3 ABLATION STUDIES

We conduct ablations experiments to analyze the contributions of the proposed serialized attention block and the spcae-channel context model. We replace each key design with a baseline counterpart or remove it from the model, and record the corresponding RD performance on DL3DV-GS. More ablation studies on context scale and positional encoding choices are presented in Appendix B.3.

**Morton Serialized Attention.** To verify the effectiveness of the serialized attention block, we conduct two variants here. The first baseline replaces all the attention blocks with MLPs. Notably, this model actually disables the space context because MLP cannot model the correlations among different Gaussians. The second baseline keeps the attention blocks while removing the Morton serialization operation. In this case, the attention is computed on a randomly permuted Gaussian sequence, which cannot guarantee that the Gaussian primitives are attended to their spatial neighbors. As shown in Fig. 7 and Table 2, the MLP backbone baseline yields significantly worse RD performance. This experiment confirms the superiority of the attention network compared to MLP.

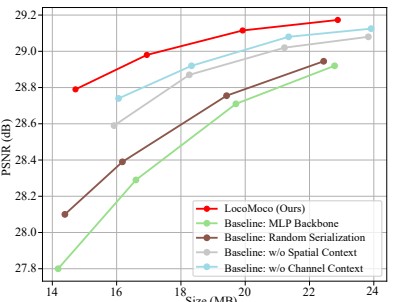

| Ground Truth Image | Original 3DGS before compression
PSNR: 26.53 dB Size: 998.87 MB | FCGS compression result
PSNR: 25.93 dB Size: 39.83 MB | Our LocoMoco compression result
PSNR: 26.15 dB Size: 37.99 MB |

Figure 6. Qualitative results on DL3DV-GS, which compares the ground truth image and the rendering results of vanilla 3DGS, FCGS, and LocoMoco. The FCGS compressor causes blurred (see blue box) and distorted (see red, black, and green boxes) reconstruction. The proposed LocoMoco achieves high-fidelity reconstruction while reducing the storage size.

| Method | BD-Rate ($\downarrow$) |
|---|---|
| LocoMoco (Ours) | 0 |
| Baseline MLP Backbone | +36.85% |
| Baseline Random Serialization | +30.62% |
| Baseline w/o Spatial Context | +18.45% |
| Baseline w/o Channel Context | +14.29% |

Figure 7. Rate-distortion curves of different ablation baselines on DL3DV-GS.

Table 2. BD-Rate to the original LocoMoco model.

Specifically, it demonstrates that the proposed attention model effectively captures the spatial dependencies, which are critical to enhance compression efficiency. The random serialization baseline also exhibits a large performance gap to the original model. It verifies that the attention model cannot perform well on weakly correlated sequences. In contrast, the attention model is effective on the Morton serialized sequence, where the spatial correlations are preserved.

**Space Context.** To validate the importance of the space context, we implement a baseline where the spatial context is removed from lossless geometry, lossy color, and lossless color compression networks. Specifically, this model predicts the symbol distribution based on hyperpriors and channel contexts. As illustrated in Fig. 7 and Table 2, applying the spatial context modeling leads to a BD-Rate gain of 18.45%. This experiment verifies that the space context is quite informative, and the proposed space context model successfully utilizes these strong correlations.

**Channel Context** Subsequently, we remove the channel context from the model while preserving the hyperprior and spatial references. Similarly, the channel context is removed from all networks. Results indicate that the channel context contributes to a significant BD-Rate improvement of 14.29%, which demonstrates the effectiveness of the channel context model. In summary, both space and channel context is important to yield the satisfactory compression efficiency.

# 6 CONCLUSION

In this work, we propose LocoMoco, an efficient feed-forward 3DGS compression method that leverages the informative long-range dependencies within large-scale 3D scenes. To this end, we first propose a large-scale context structure tailored for the 3DGS data using Morton serialization. Its available context capacity reaches 1024 Gaussian primitives, which has been significantly expanded compared to existing voxel-based methods. Furthermore, we devise an attention architecture to effectively model the long-range dependencies. Moreover, we formulate a space-channel context model to fully exploit the contextual features conveyed by the large-scale context for precise probability density estimation. Extensive experiments demonstrate that the proposed method achieves significant improvements in compression efficiency while preserving the low-latency coding strength of the feed-forward 3DGS compressor.

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

# Appendix

## A  IMPLEMENTATION DETAILS

### A.1  MODEL ARCHITECTURE

**Lossless Geometry Compression**  The lossless geometry compression model consists of the hyper analysis transform, hyper synthesis transform, and space-channel context model. Both the analysis and synthesis transform are composed of 2 MLPs and 1 self-attention block, where the channel dimension is 16. The latent vector $\boldsymbol{u}_{\text{geo}}$ includes 8 channels. Both the space and channel context model consists of one MLP and one attention block. The channel group size is set to 1 for lossless geometry compression.

**Lossy Color Compression**  For lossy color compression, the architectures of the transform coding network and context model are consistent with the lossless compression pipeline. Since the lossy compressor needs to store more information in the latent embedding, the dimension of features and latent $\hat{\boldsymbol{y}}$ is set to 128 here. The group size of the channel context model is $[32, 32, 64]$. The hyperprior embedding $\hat{\boldsymbol{z}}$ includes 32 channels.

**Lossless Color Compression**  The transforms and context model of the lossless color compression model share the same architecture as the ones in geometry compression. Channel dimension is set to 48 here. The latent embedding $\boldsymbol{u}_{\text{color}}$ has 48 channels, and the channel group size is $[8, 8, 16, 16]$. In particular, we predict the $\hat{\boldsymbol{c}}_{\text{lossless}}$ based on the decoded color attributes $\hat{\boldsymbol{c}}_{\text{lossy}}$ by connecting them in the same sequence $\hat{\boldsymbol{c}}$. Then, we perform Morton serialization on the concatenated sequence and use an self-attention block to introduce features from $\hat{\boldsymbol{c}}_{\text{lossy}}$ to predict $\hat{\boldsymbol{c}}_{\text{lossless}}$.

**DGCNN**  The attention block in LocoMoco employs a DGCNN (Wang et al., 2019) to produce the positional encoding. The $k$-th DGCNN layer can be described as:

$$\boldsymbol{f}_i^{k+1} = \sigma_k([\boldsymbol{f}_i^k, \boldsymbol{f}_i^k - \boldsymbol{f}_j^k]), \quad \text{for } (i, j) \in \mathcal{E}, \tag{13}$$

where $\sigma_k$ is a MLP, $\boldsymbol{f}_i^k$ is the feature of $i$-th Gaussian, $\mathcal{E}$ is a graph where each Gaussian is connected to its $K$ nearest neighbors in the feature space. After each layer, $\mathcal{E}$ is updated by recomputing the nearest neighbors in the new features. The first DGCNN layer receives the quantized Gaussian positions as input, i.e., $\boldsymbol{f}_i^0 = \hat{\boldsymbol{\mu}}_i$. After several layers, DGCNN applies a maximum pooling layer to extract the global features, and concatenates global and MLP features to produce the final embedding. In LocoMoco, we build the positional encoding module with a 3 DGCNN layers.

**Division Network**  The division network, which is employed to split Gaussians for lossy and lossless compression, includes only 1 MLP and 1 attention block. Its channel dimension is set to 32.

### A.2  TRAINING RECIPE

In this section, we introduce the training pipeline of LocoMoco. To stabilize the training process, we adopt a multi-stage training pipeline to optimize the LocoMoco model, where we firstly train the framework components and then perform end-to-end tuning. As an initialization, the lossy color compression model and division networks are trained with a loss function as:

$$\mathcal{L} = R + \lambda D(\delta(\boldsymbol{G}), \delta(\tilde{\boldsymbol{G}})) + w * \frac{N_{\text{lossless}}}{N}. \tag{14}$$

With this training objective, we are able to vary the lossy-lossless division ratio by adjusting the hyperparameter $w$. We train these networks with targeted lossy compression ratio as $[85\%, 80\%, 75\%, 70\%, 65\%]$. This strategy allows a separate training of the lossy color transform coding without reliance on the final $\lambda$. For the lossless compression part, we train the entropy model with the rate loss depicted in Eq. 10 and Eq. 12 with the quantization steps initialized from FCGS. Besides, we need to train the lossy and lossless color compression model sequentially, because the prediction of $\hat{\boldsymbol{c}}_{\text{lossless}}$ relies on the lossy compression results $\hat{\boldsymbol{c}}_{\text{lossy}}$. During the lossless color model training, the lossy color model and the division model are fixed. After the individual training, we

further fine-tune all modules in an end-to-end manner with the full loss in Eq. 9, where all model parameters and the learnable quantization steps are tuned, thereby achieving a global rate-distortion optimization.

# B  SUPPLEMENTARY EXPERIMENTAL RESULTS

## B.1  FIDELITY METRICS

In the main paper, we provide the rate-distortion performance in terms of data size and PSNR. Here we provide additional fidelity metrics evaluation, including SSIM and LPIPS. Our method achieves superior rate-distortion performance, maintaining a similar fidelity with vanilla 3DGS while reducing the storage overhead. Compared with FCGS, our method yields high visual quality with a reduced storage overhead.

Table 3. Additional evaluation results in terms of PSNR, SSIM, LPIPS and the data size.

| Method | PSNR (dB) ↑ | SSIM ↑ | LPIPS ↓ | Size (MB) ↓ |
|---|---|---|---|---|
| Vanilla 3DGS | 29.344 | 0.8983 | 0.147 | 372.50 |
| FCGS ($\lambda = 0.0001$) | 29.178 | 0.8966 | 0.149 | 27.96 |
| FCGS ($\lambda = 0.0016$) | 28.692 | 0.8896 | 0.155 | 15.85 |
| LocoMoco ($\lambda = 0.0001$) | 29.186 | 0.8968 | 0.149 | 26.07 |
| LocoMoco ($\lambda = 0.0016$) | 28.789 | 0.8901 | 0.156 | 14.72 |

## B.2  STORAGE PROPORTION ALLOCATION

Fig. 8 illustrates the storage proportion of each component in our compression result, across various rate-distortion trade-offs. With the hyperparameter $\lambda$ descending, the model tends to pose more emphasis on the low distortion aspect, and the storage proportion of the lossless color grows, which facilitates obtaining a higher representation quality.

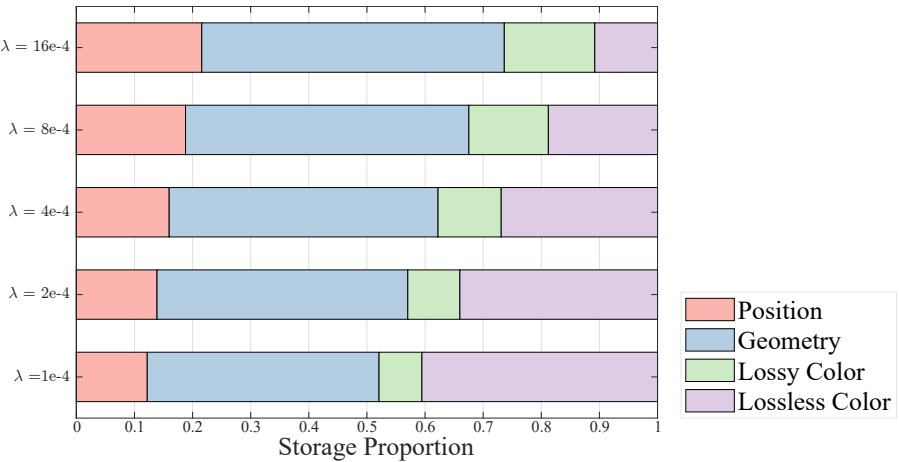

Figure 8. Proportion of each component's storage in LocoMoco, evaluated on the DL3DV-GS dataset. The division mask size, though included in the total file size, represents less than 1% of the final result and is not displayed in this figure.

Table 4. BD-Rate to the original LocoMoco model.

| Method | LocoMoco (Ours) | Context-512 | Context-128 | Context-16 | Baseline: Sinusoidal PE |
|---|---|---|---|---|---|
| BD-Rate (↓) | 0 | + 4.05% | + 7.97% | + 27.18% | + 7.93% |

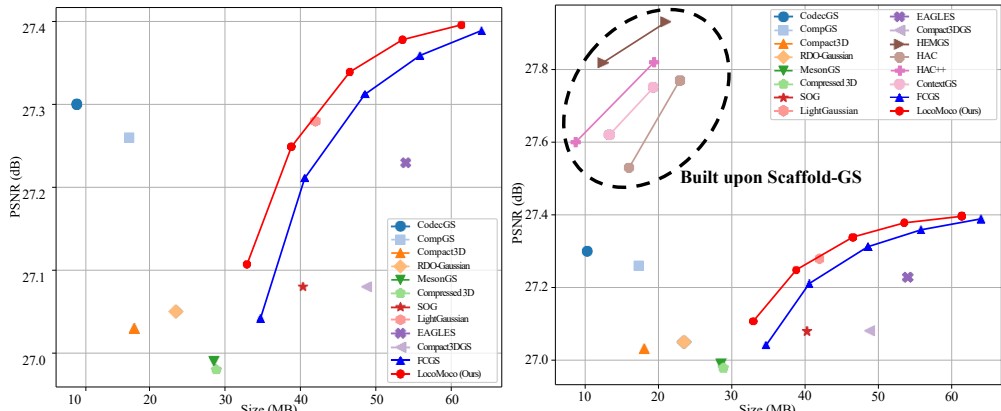

Figure 9. Rate-distortion performance comparison including the per-scene per-train compression baselines. The right figure includes the baselines built upon Scaffold-GS (Lu et al., 2024), which achieve superior performances and distinguish themselves in the RD illustration.

## B.3 EXTENSIVE ABLATION STUDIES

**Influence of the Context Window Size.** Context modeling serves as a critical role in compression tasks, contributing to both the transform coding and entropy model. We conduct ablation on the context capacity where the context window length $L$ is set to 512, 128 and 16. Recall that in the original LocoMoco model, the context size is 1024. As shown in Tab. 4, reducing the context window size results in performance degradation. Specifically, reducing the context window size to 512 leads to less than 5% performance drop. Smaller context windows fail to capture the data dependencies and cannot provide satisfactory compression results. These results confirm the significance of our main idea as fully utilizing the data dependencies by introducing long-context modeling for 3DGS compression.

**Effectiveness of the DGCNN Positional Encoding.** Positional encodings contribute significantly to attention modules and Transformer blocks. To evaluate the effectiveness of our DGCNN-based positional encoding, we replace all of the DGCNN layers in our attention modules with plain sinusoidal positional encoding (Vaswani et al., 2017). As shown in Tab. 4 Baseline: Sinusoidal PE, this variant leads to a 7.93% RD performance degradation relative to our proposed model. This verifies the DGCNN's effectiveness to the compression pipeline. Compared to plain sinusoidal positional encoding, DGCNN dynamically captures the local and global geometric information and contributes to an effective attention module.

## B.4 COMPARISON WITH PER-SCENE PER-TRAIN COMPRESSION METHODS

For the sake of completeness, we provide the rate-distortion performance comparisons against the per-scene per-train baselines in Fig. 9 and Table 5. We compare LocoMoco with a wide range of baselines here, including HEMGS (Liu et al., 2024a), ContextGS (Wang et al., 2024b), HAC (Chen et al., 2024), HAC++ (Chen et al., 2025b), CodecGS (Lee et al., 2025), CompGS (Liu et al., 2024b), Compact3D (Navaneet et al., 2024a), RDO-Gaussian (Wang et al., 2024a), MesonGS (Xie et al., 2024), Compressed3D (Niedermayr et al., 2024), SOG (Morgenstern et al., 2023), LightGaussian (Fan et al., 2023), EAGLES (Girish et al., 2024), Compact3DGS (Lee et al., 2024), and FCGS (Chen et al., 2025a).

Please note that among all these methods, only FCGS (Chen et al., 2025a) and LocoMoco are generalizable feed-foward codecs. It is natural that the per-scene per-train methods achieve higher RD performance, since these methods overfit to each scene while the feed-forward methods learn towards a generalizable compression capability. However, as shown in Table 5, these methods require an exhaustive optimization for compressing each 3DGS sample, which greatly slows down the coding speed. In contrast, feed-forward 3DGS codecs support much faster processing, because of the computational overhead of network inference is much lower than the one of training.

Table 5. Time consumption of different 3DGS compressors. For feed-forward methods, both the encoding and decoding time are counted.

| Method | Time for Compression (s) | Method | Time for Compression (s) |
|---|---|---|---|
| LightGaussian | 978 | HAC | 1800 |
| Compact3D | 1297 | ContextGS | 5000 |
| Compressed3D | 873 | HEMGS | 3600 |
| SOG | 1068 | HAC++ | 2292 |
| EAGLES | 518 | FCGS | 26.98 |
| Compact3DGS | 938 | LocoMoco (Ours) | 24.22 |

## B.5 VISUALIZATIONS

As an extension of Fig. 1, we include the correlation map between the Gaussians in a context window in Fig. 10. We also illustrate another example of the scene *ficus*, where strong correlations are observed in the region of leaves. These visualization intuitively verifies the significance of including long-context modeling in 3DGS compression pipelines. We also include extensive visualization results in Fig. 11.

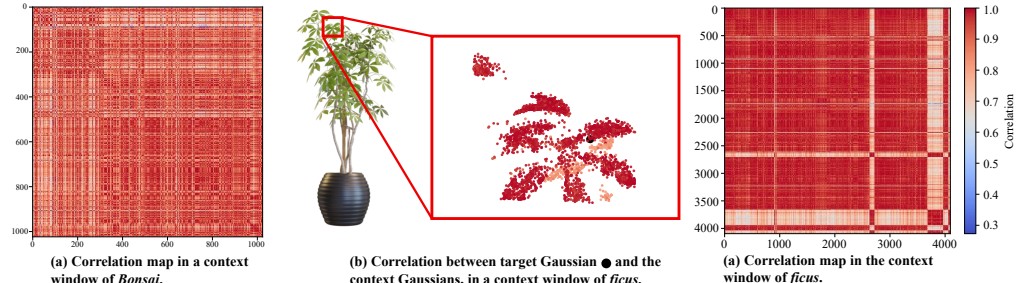

(a) Correlation map in a context window of *Bonsai*.  (b) Correlation between target Gaussian ● and the context Gaussians, in a context window of *ficus*.  (a) Correlation map in the context window of *ficus*.

Figure 10. (a) Correlation map between the Gaussians in one context window of scene *Bonsai*. (b) Correlation between one target Gaussian and the others in a same context window in scene *ficus*. (c) Correlation map between the Gaussians in one context window of scene *ficus*.

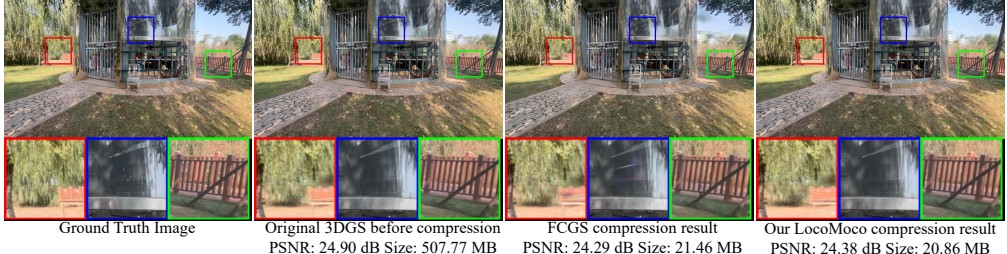

Ground Truth Image    Original 3DGS before compression    FCGS compression result    Our LocoMoco compression result
                       PSNR: 24.90 dB Size: 507.77 MB      PSNR: 24.29 dB Size: 21.46 MB    PSNR: 24.38 dB Size: 20.86 MB

Figure 11. Qualitative results on DL3DV-GS, compared with the ground truth image and the rendering results of vanilla 3DGS, FCGS, and LocoMoco. Our proposed LocoMoco achieves high-fidelity reconstruction while reducing the storage size.

## C THE USAGE OF LARGE LANGUANGE MODELS (LLMs)

During the preparation of this manuscript, large language models (LLMs) were utilized solely as writing assistants to refine language expression, including improvements in clarity, grammar, and readability. All substantive intellectual contributions, encompassing the conception of research ideas, the development of methodologies, and the design of experiments, were carried out exclusively by the authors without the involvement of LLMs.

