# OpenReview forum: "Feed-Forward 3D Gaussian Splatting Compression with Long-Context Modeling"
_ICLR.cc/2026/Conference — ICLR 2026 Conference Withdrawn Submission_

### Official Review · Reviewer_a9Mf · 2025-10-28

**Soundness:** 2
**Presentation:** 1
**Contribution:** 2
**Rating:** 2
**Confidence:** 5

**Summary:**

This paper provides a feed-forward 3DGS compression framework with long-range context modeling: LocoMoco. The 3DGS is transferred to a 1D Gaussian primitive sequence by Morton sorting and split into independent patches. For each patch, attention-based positional embedding and spatial-channel domain auto-regressive modules are proposed. Experimental results show that LocoMoco outperforms FCGS with about 10% BD-Rate gain and lower time complexity.

**Strengths:**

1. The paper provides abundant subjective and objective evaluations and thorough ablation studies.
2. The proposed method achieves about 10% BD-Rate gain compared with FCGS while maintaining excellent parallelism.

**Weaknesses:**

1. Clarity of the model

The proposed method exhibits a high degree of overlap with FCGS. Therefore, exact implementation differences and the precise source of the reported gains must be explicitly clarified:

(a) Context window splitting:

3D Morton code suffers from discontinuity: consecutive Morton codes can map to voxels that are far apart in Euclidean space. How does the proposed method solve this problem?

(b) DGCNN and the attention module:

In Fig. 4 left and Supplement A.1, the DGCNN block takes only quantized positions as input and outputs a positional embedding. It seems that the following self-attention and MLP modules in Fig.4 left take only the positional embedding as input, rather than latent features from color/geometry attributes, which is confusing. The authors should clarify how the positional embedding combines with latent features from color/geometry attributes.

(c) Analysis and synthesis transforms:

Please provide the full architectural detail. Specifically, clarify how the two MLPs and the single attention module are cascaded, e.g., whether the MLPs precede or follow attention, where residual connections are placed, and how channel dimensions are matched.

(d) Division Network.

In Supplement A.1, the sentence "Its channel dimension is set to 32" is ambiguous. Please specify whether this refers to the hidden feature channels or the output mask channels. Additionally, clarify: Is the mask a single binary decision per Gaussian primitive, or is a separate decision made for each color channel of every primitive? How is the mask entropy-coded and signaled in the bitstream? And is the mask shared for all the 1024-gaussian context windows?

(e) Computational complexity analysis and parallelism:

The authors should provide a detailed explanation of the lower time complexity compared with FCGS. Specifically, it is necessary to clarify how the independent and parallel processing of the context window affects time complexity. It is better to report the proportion of the encoding/decoding time consumed by each major module (context window partition/grid generation, transform, entropy model, etc.) for both LocoMoco and FCGS.

(f) Ablation on context window size

Experiments are missing for window sizes larger than 1024, and please report how the window size affects encoding/decoding time, memory allocation, except for compression efficiency.

2. Writing:

Figure 4 is too large.

In Supplement A.1, the description of how the lossy color branch guides the lossless color coding, as well as the explanation of the DGCNN module, is better to be moved into the main text.

**Questions:**

See weaknesses.

---

### Official Review · Reviewer_jL4W · 2025-10-29

**Soundness:** 3
**Presentation:** 2
**Contribution:** 2
**Rating:** 2
**Confidence:** 4

**Summary:**

This paper introduces LocoMoco, a feed-forward compression framework for 3D Gaussian Splatting (3DGS) that eliminates the need for per-scene optimization and ground-truth multi-view images, while maintaining high compression efficiency. The method leverages Morton serialization to enable long-range context modeling. It employs an attention-based transform and a space–channel autoregressive entropy model to capture inter- and intra-correlation of Gaussian primitives effectively. Experiments demonstrate that LocoMoco achieves superior rate-distortion performance on commonly used real-world datasets compared to previous feed-forward 3DGS compression models like FCGC.

**Strengths:**

1. Morton serialization: Introduces Morton-order sorting to enable long-range spatial dependency modeling among thousands of Gaussian primitives while maintaining spatial locality.
2. Space–channel autoregressive entropy model: Proposes a fine-grained space-channel context modeling strategy that jointly captures inter- and intra-correlations of Gaussian primitives.

**Weaknesses:**

1. Limited novelty beyond integration: The approach mainly combines Morton serialization, attention-based transform coding, and autoregressive entropy modeling, all known components in point cloud compression, with limited theoretical innovation.
2. Computational and memory overhead: While the attention-based transform coding expands the receptive field, the paper does not quantify the added GPU memory or computational cost compared to simpler MLP-based feed-forward designs.
3. No evaluation with Gaussian pruning approaches (eg, Compact3D, LightGaussian, Trimming): Since pruning reduces local correlations in long sequences, it is unclear whether the proposed long-context modeling still offers consistent gains over FCGS under sparse or irregular Gaussian distributions.
4. Unclear figure/description: In Fig. 3, it is unclear how the decoded lossy color serves as a condition for lossless color compression, as the conditioning mechanism is not explained in the text.

**Questions:**

Limited ablation on sorting strategy: The ablation compares only with random sorting; could the authors evaluate other ordering methods (e.g., Hilbert or octree) to demonstrate the advantage of Morton serialization more convincingly?

---

### Official Review · Reviewer_24Cv · 2025-10-30

**Soundness:** 3
**Presentation:** 4
**Contribution:** 3
**Rating:** 6
**Confidence:** 3

**Summary:**

This paper presents a feed-forward Gaussian compression framework that effectively models the long-range spatial dependencies of 3D Gaussians. To model long contexts, it introduces a transform coding network that applies self-attention on Morton-serialized context windows. Moreover, the space-channel context modeling enables full leverage of informative contexts, leading to improved reconstruction fidelity. Experimental results demonstrate that it outperforms the BD-rate of the existing feed-forward compression method with grid-based context modeling.

**Strengths:**

- It indicates a meaningful technical limitation of the prior feed-forward 3DGS compression method, the limited receptive field of the transform coding network, and addresses it by designing an architecture for modeling long contexts.

- The proposed fine-grained space-channel context model enables more accurate probability distribution estimation, resulting in improved BD-rate.

- The manuscript is well organized to follow the logical flow, while the evaluations successfully validate the contribution of each component.

**Weaknesses:**

- Despite fast compression due to a feed-forward framework, several optimization-based compression methods achieve better BD-rate performance.

- The training process demands a large-scale 3DGS dataset, which incurs substantial computational costs associated with optimizing thousands of 3D scenes.

- This method does not achieve a noticeable improvement in compression speed, which is also an important factor for feed-forward compression.

- The feed-forward compression may require excessive GPU memory for training and inference.

**Questions:**

- Could you explain the training costs for training the network? It may require extensive computation time and multiple high-end GPUs for optimizing numerous 3D scenes..

- Also, could you explain the memory consumption for inference? The MLP computation for a large number of Gaussians may require substantial GPU memory and could limit scalability when applied to large-scale scenes (e.g., city-scale).

---

### Official Review · Reviewer_G4as · 2025-10-31

**Soundness:** 3
**Presentation:** 3
**Contribution:** 3
**Rating:** 6
**Confidence:** 4

**Summary:**

This paper introduces LocoMoco, a novel feed-forward compression method for 3DGS that effectively models long-range spatial dependencies. The authors address the limitations of existing feed-forward compressors, which struggle with limited receptive fields and inadequate context modeling. The proposed approach leverages Morton serialization to organize Gaussians into large-scale context windows, uses self-attention-based transform coding to extract informative latent representations, and employs a fine-grained entropy model for accurate probability estimation. The method achieves  state-of-the-art performance among feed-forward codecs.

**Strengths:**

1. The paper effectively addresses long-range dependencies by introducing a large-scale context structure. This is a significant improvement over voxel-based local contexts.

2. The combination of Morton serialization, self-attention transform coding, and space-channel context modeling is well-motivated. The design ensures spatial coherence and enables efficient information aggregation across distant Gaussians.

3. The method demonstrates consistent improvements across multiple datasets (DL3DV-GS, Mip-NeRF 360, Tanks & Temples) in both quantitative metrics (PSNR, BD-Rate) and qualitative visualizations. The ablation studies convincingly validate each component.

**Weaknesses:**

1. Shortcomings still exist in the experiments. They primarily include the lack of comparison on computational complexity and the lack of ablation studies on the design choice of using both lossy and lossless color compression simultaneously.

2. The paper lacks discussion on training overhead and hardware requirements

**Questions:**

The paper primarily compares the impact of reducing the context window size on performance. Would further increasing the window size lead to additional improvements in model performance?

---

### Note · Authors · 2025-11-13

I have read and agree with the venue's withdrawal policy on behalf of myself and my co-authors.